# Nutrient Film Technique-Based Hydroponic Monitoring and Controlling System Using ANFIS

Vito Vincentdo [1] and Nico Surantha [1,2,*]

---

[1] BINUS Graduate Program—Master of Computer Science, Computer Science Department, Bina Nusantara University, Jakarta 11480, Indonesia

[2] Department of Electrical, Electronic and Communication Engineering, Faculty of Engineering, Tokyo City University, Setagaya-ku, Tokyo 158-8557, Japan

[*] Correspondence: nico.surantha@binus.ac.id

**Abstract:** Most people are now aware of the importance of a healthy lifestyle, including the importance of consuming vegetables. As a result, the demand for vegetables has increased, and so their production needs to be increased. Currently, most plantations use soil as a growing medium, which is time-consuming and requires a significant amount of space. To modernize cultivation, hydroponic techniques should be adopted. However, implementing hydroponics can be challenging as it requires precise pH and nutrient adjustments. The previous research has proposed the hydroponic pH and nutrient control using the Sugeno fuzzy method. However, in Sugeno fuzzy method, there is no systematic procedure in designing the fuzzy controller, thus, the design relies on hydroponic expert knowledge. To address this issue, a smart hydroponic system was developed using the adaptive neuro-fuzzy inference system (ANFIS) method, which allows for automatic adjustments based on the collected dataset and remote control through internet of things (IoT) technology. This study showed that the system could accurately adjust pH and nutrient levels, allowing plants to grow better. Furthermore, the fuzzy controller created using ANFIS is 67% more accurate than creating the fuzzy controller using the Sugeno fuzzy method. Finally, the web application dashboard of the proposed system is also presented in this paper.

**Keywords:** automatic hydroponic system; internet of things; ANFIS; agriculture; fuzzy logic control; artificial intelligence





## 1. Introduction

In this modern era, people have come to understand the importance of health, and the implementation of a healthy lifestyle has become prevalent. One of the popular healthy lifestyles involves the consumption of healthy food. Vegetables are highly regarded as a healthy food due to their beneficial substances for the human body. These include vitamins, minerals, fiber, and phytochemical compounds. These substances make sufficient vegetable consumption become a solution to prevent various diseases [1]. As the public realizes the importance of a healthy lifestyle, the demand for vegetable production increases. However, in Indonesia, vegetable plantation is still performed traditionally by using soil as the medium. This method needs to be modernized because it requires a large space and has a slow pace of production [2]. A form of modernization that can be implemented is by planting using hydroponic technique.

Hydroponics is a planting process that utilizes water as the growing medium. When plants are cultivated using hydroponics, they can grow faster and larger compared to traditional growing methods due to the provision of nutrients to obtain maximum crop yields [3]. However, implementing this technique requires water reservoirs with precisely calibrated nutrient levels, which can be seen as an obstacle for some people who wish to grow crops using hydroponics, as it requires a bit of complexity in the setup.

Several studies have tried to resolve the difficulty of nutritional adjustment techniques. In a study by Mehra et al. [4], the deep neural network (DNN) was implemented to control the environmental conditions of the hydroponic system. There are also researchers who proposed an automatic hydroponic system by using Bayesian networks [5]. Another research that intrigued the authors of this study was an automatic hydroponic system that implemented fuzzy logic [6]. According to Denai et al. [7], controlling a system by using fuzzy logic results in a good performance and it only needs a simple mathematical model to formulate the algorithm.

However, a weakness found in fuzzy logic was the absence of a systematic procedure for designing a fuzzy controller. This is where the adaptive neuro fuzzy inference system (ANFIS) plays an important role in overcoming this issue, as ANFIS can automatically perform parameter adjustments [8]. Researchers have used ANFIS to train the fuzzy controller using real data from existing problems. In this study, we present an internet-of-things (IoT) and ANFIS-based hydroponic control system. The IoT system is designed to monitor and regulate the pH and nutrient levels of the plants being observed. Then, the ANFIS algorithm is subsequently used to regulate the pH-up, pH-down, water, and nutrition pumps.

The main contribution of this paper are as follows:

- A prototype of indoor IoT-based nutrient film technique hydroponic control system has been developed, which integrates sensors connected to both Arduino and Raspberry Pi 4;
- Implementation of an ANFIS model to provide precise control actions based on the parameters measured in a hydroponic system;
- Performance comparison between the implementation of FIS and ANFIS in creating a smart hydroponic system that can automatically control the nutrient and pH level in the system.

The remaining sections of this study are organized as follows. Section 2 discusses the related studies on the methods of a smart hydroponic system. Section 3 explains the methodology, defining the proposed method, and the design prototype in order to achieve the aim of the research. Section 4 discusses the evaluation after the system has successfully been created. The authors tested and evaluated the system to ensure that the goals of this research are achieved. Section 5 contains the conclusions and suggestions for future work.

## 2. Related Works

This section discusses the research conducted on an intelligent IoT-based hydroponic system that incorporates both IoT and machine learning technologies. It also explores related studies on the proposed method. Furthermore, we will review various literature sources related to the development of this system, which will be discussed in detail below.

### 2.1. Related Works in Hydroponics IoT

There have been numerous past studies on the development of intelligent IoT-based hydroponic systems. For instance, research conducted by Gomes et al. [9] performed a comprehensive overview of time-sensitive applications in fog computing environments to understand the current status and approach toward the real-time concept. As a result, the near real-time concept is being used in approaches such as big data, IoT, and remote sensing, reinforcing that these environments tend to not respond in hard real-time due to their present characteristics. Continuing the research on the near real-time concept for IoT, Tran et al. [10] have proposed a hydroponic farming framework that leverages fog computing to provide low-cost data collection and innovative data analysis methods for creating intelligent farming systems. The framework collects data from multiple IoT sensors in the garden, which is then filtered and analyzed by artificial neural network models deployed in the fog landscapes. The models are trained using a vast amount of historical farming data in the cloud. The developed prototype demonstrates the effectiveness and performance

of the proposed approach, showing that it is practical and ready for implementation in real-world scenarios.

Ibayashi et al. [11] designed a smart hydroponic system by using a wireless sensor network (WSN). This study discusses the usage of 400 MHz wireless band to control hydroponic systems where the normal frequency that is generally used is 2.4 GHz. By using a lower frequency, the wavelength will also be longer. The longer the wave, the better the obstacle diffraction will be. The parameters measured are wavelength, temperature, and humidity. Inside this system, the actuator works in accordance with the evapotranspiration level that the user specifies, and it can be calculated by computing the vapor pressure deficit (VPD) and relative light intensity (RLI). Suppose the evapotranspiration level surpasses the predetermined level, in that case, the actuator will trigger the program and send an infrared signal to the remote control to turn on the nutrition liquid pump.

Mehra et al. [4] discuss the implementation of deep neural networks (DNN) in the deep water culture smart hydroponic system. The input parameters for DNN are pH, water level, temperature, light intensity, and humidity. These parameters will be computed through a model trained on the cloud and provide a classification of the required actions to optimize the hydroponic environment. The results from the system can only classify which actuators have to be turned on or off. In the abovementioned smart hydroponic system, all of the research uses a sensor to get the required data to be processed further. As the number of devices increases, it becomes necessary to implement methods to choose the most appropriate sensors for each task. This approach is critical in applications with low latency requirements, so a process to identify the most reliable sensor that delivers accurate data is important. Costa et al. [12] presented a method called greatest-of-actual-time (GoAT) to rank sensors based on the concept of active perception. The solution was tested using four real datasets. The results showed that GoAT provides a reliable utilization of sensor data and requires low computational resources while reducing latency in the sensor selection process.

Alipio et al. [5] implemented the nutrient film technique (NFT) hydroponic technique, which continuously circulates nutritional water from the reservoir to the planting medium. The water flows through the gutter and passes through all of the plants' roots. This study utilized a Bayesian network to automatically adjust the pH and EC in the reservoir. The pH, EC, humidity, light intensity, and water temperature were used as input parameters for the Bayesian network. Once all the required sensor data had been collected, the Bayesian network processed it and issued a command to the actuators to automatically adjust the hydroponic system's environment. All of the data were sent to the cloud, allowing it to be monitored via the web.

According to Herman and Surantha [13], implementing fuzzy logic to control actuators can lead to interesting outputs, as it not only turns the actuators on or off but also predicts the duration that they need to be activated. The hydroponic deep water culture (DWC) technique was utilized in this study, which is the simplest hydroponic technique requiring only a water reservoir, with the plants placed directly above it.

The proposed approach in this research is to implement ANFIS to control the actuators in a hydroponic system. At this moment, there has not been specific research on implementing ANFIS to control both pH and dissolved nutrient levels in a hydroponic system. ANFIS is capable of independent learning based on the input-output pairs provided, enabling it to automatically create a fuzzy controller based on the data. This reduces the dependence on expert knowledge required for traditional fuzzy control. This study also includes a comparison of the accuracy between ANFIS and fuzzy Sugeno. The performance of both methods will be observed when confronted with the exact same environment. Table 1 summarizes the advantages and disadvantages of previous research studies in hydroponic IoT.

**Table 1.** Related works in hydroponic IoT.

| Research | Measured Data | Method | Advantage | Disadvantage |
|---|---|---|---|---|
| Mehra et al. [4] | Temperature, humidity, water level, LDR, PPM | Deep neural network | The architectural implementation of edge-fog-cloud computing | During the training phase, the pH is mentioned but does not appear in the testing phase; instead, PPM unexpectedly appears |
| Ibayashi et al. [11] | Wavelength, temperature, humidity | WSN | By using wireless for each sensor node, the complexity of wiring for the overall system can be reduced | In research, only a few nodes are utilized; the use of a large number of sensors may impact signal transmission |
| Alipio et al. [5] | Level pH, electrical conductivity, relative humidity, light intensity, water temperature | Bayesian network | The research uses large amounts of datasets that lead to good performance results by the created model | The Bayesian network is quite complex and might need larger computing resources; therefore, a comparison with other methods is needed |
| Yolanda et al. [6] | pH, EC | Fuzzy | The proposed method has a tolerance limit of approximately 15 min to complete the process and attain the desired value; based on testing, the system consistently brings the EC and pH values to the desired value within no more than 15 min | Lack of results from plants grown using the system |
| Herman and Surantha [13] | EC, pH, water level, humidity | Fuzzy | A comparison between planting using the proposed method and traditional methods show that the plants grown using the proposed method grow better | The accuracy of the system is not presented |
| Gomes et al. [9] | - | Various | Provides a state-of-the-art research on near real-time fog computing | - |
| Tran et al. [10] | IoT sensors | ANN | The fog computing approach enables the updating of intelligent models in real-time while reducing communication costs and response time | The effectiveness and feasibility of the proposed approach may vary depending on the specific environmental and farming conditions |
| Costa et al. [12] | Humidity, temperature, light, and voltage values | Gradient tree boosting class, fuzzy | Reducing the heavy performance that needs to be analyzed by the computing resources | Applicable if using many sensors |
| Proposed | TDS, pH, water level | ANFIS | Comparing ANFIS and FIS Sugeno in the hydroponic sector with multiple inputs, ANFIS performs better in controlling the system; furthermore, the accuracy of ANFIS in controlling the system is present | - |

*2.2. Related Research Using ANFIS*

ANFIS is a method that works well for controlling actuators in situations where there is uncertainty or non-linearity in the system being controlled. It can learn to adjust its parameters based on input data and optimize its control actions to achieve the desired output. An example of implementing ANFIS to control such a system is the research

conducted by Kaiser et al. [14] that implemented the ANFIS method to move a wheelchair. In this study, the input parameters were extracted from the electromyogram (EMG) signal that was captured by the sensor, then the input data were classified by the rule based on each hidden layer, and the result of the classification was an action to move the wheelchair. Based on the evaluation results, the ANFIS method achieved a high accuracy of 96.85%.

The authors in research [9,10,13–15], used ANFIS and IoT to improve crop production in greenhouses. In this system, four key weather parameters, such as temperature, humidity, sunlight, and soil moisture are continuously monitored by sensors and transmitted as input variables to the fuzzy control system. The fuzzy control system then processes this data with the help of ANFIS to generate predictions for optimal weather conditions. These predictions provide farmers with valuable information to determine their crops' most suitable temperature and humidity levels. The results showed improved learning efficiency and prediction accuracy, making the ANFIS-based smart greenhouse a feasible solution for automated maintenance.

Jorge et al. [16] utilized ANFIS to train a fuzzy model for estimating the water evapotranspiration of tomato plants in a greenhouse. The model employed two variables, namely VPD and plant mass. Based on the observation performed for a period of two days with a fully saturated plant and no irrigation, the model yielded an error rate of 4.48%. The proposed model reduced the number of variables used and provided cost savings for protected agriculture. Another interesting study by Arora and Keshari [17] focused on finding the best ANFIS models for predicting dissolved oxygen levels in water for assessing water quality. The study used two types of models: ANFIS-GP and ANFIS-SC. Four different models were developed and evaluated using the root mean square error and the coefficient of determination (R2). The results showed that both ANFIS models were effective in predicting dissolved oxygen levels, with ANFIS-GP performing better with an R2 of 0.953 compared to 0.911 for ANFIS-SC.

Another research study by Ali et al. [18], which also focused on dissolved oxygen, implemented ANFIS to design a control system for measuring fluid flow in industrial processes. Specifically, the system regulates water flow rate at a predetermined set point using a PID-based system with an electric ball valve as an actuator. The study compared five different control models, including a standard PID method, PID-auto-tuning method, PID-FA method, and ANFIS-PID-FA method, and found that the ANFIS-PID-FA model was the best performing control model. The researchers utilized data from the PID-FA training as input for ANFIS, which helped to achieve the best results. Lastly, in the agricultural sector that used an ANFIS method by [19], the authors developed a systematic approach using FIS, ANN, and ANFIS models to evaluate the technical performance of irrigation networks, considering uncertainties in the water exploitation process. Results showed that the ANFIS model was more accurate in predicting adequacy, efficiency, and equity indicators with lower errors compared to the FIS model, and the study suggests that ANN and ANFIS models can be used as useful guidelines to evaluate the performance of agricultural water distribution systems, considering uncertainties in the process.

The studies mentioned above suggest that the agricultural sector can benefit from IoT technology through the use of well-designed hardware and sensors. The research conducted by Herman and Surantha [13] inspired this study, as their use of fuzzy logic showed promising results. However, a weakness of fuzzy logic is the lack of a systematic procedure for designing a fuzzy controller, which often relies on expert knowledge. To improve on this weakness, a more advanced fuzzy logic technique, such as ANFIS, can be implemented.

On the contrary, ANFIS can automatically design a fuzzy controller based on an accurate dataset. In previous studies, ANFIS was used for controlling greenhouse [15], for estimating the water evapotranspiration of a tomato plant in a greenhouse [16], and for testing which ANFIS model would perform better in checking dissolved oxygen in water [17]. However, ANFIS has not been used for controlling pH and TDS (total dissolved solids) in hydroponic systems before. Therefore, the objective of this study is to implement

ANFIS to control pH and TDS in a hydroponic system. The performance of the ANFIS model will be compared with Sugeno fuzzy to determine which method is more accurate in controlling the hydroponic system. Table 2 summarizes the previous research studies related to ANFIS.

**Table 2.** Related research using ANFIS.

| Research | Measured Data | Method | Application | Advantage | Disadvantage |
|---|---|---|---|---|---|
| Kaiser et al. [14] | MAV, RMS, SSC, ZC, LM | ANFIS | Digital health | Compares the cost of creating a wheelchair using the proposed method with other publicly available wheelchairs and shows that the proposed method is much more cost-effective | Examples of the dataset and the total number of items in the dataset are not displayed |
| Kirci et al. [20] | Temperature, humidity, sunlight and soil-moisture | ANFIS, IoT | Greenhouse agriculture | The security of IoT data is addressed and a winter simulation is included | Lack of generalizability to other contexts, potential cost and complexity of implementation |
| David et al. [16] | Relative humidity, vapor pressure deficit | ANFIS | Greenhouse agriculture | The proposed system can help reduce the number of variables required for ET estimation | The experiment results only cover two days and a longer timeframe is needed to assess the stability of the proposed system |
| Arora and Kehari [17] | Temperature, biological oxygen demand (BOD), COD, conductivity and ammonia | ANFIS | Water quality | Comparing ANFIS grid partitioning and ANFIS subtractive clustering | Research only focuses on the Delhi stretch of the Yamuna River in India |
| Ali et al. [18] | PID-FA training data | ANFIS | Industrial | The use of the ANFIS-PID-FA model results in the smallest overshot and undershot on the water level and output flow | The study is based on simulation results rather than real-world data, so it is unclear how well the proposed system would perform in a practical setting |
| Sharifi et al. [19] | Delivered water | FIS, ANN, ANFIS | Agriculture | Comparing FIS, ANN, and ANFIS, ANFIS has the best performance | The research only considers one input (water distribution) while neural network type of method works best with multiple inputs |
| Proposed | TDS, pH, water level | ANFIS | Hydroponic | Comparing ANFIS and FIS Sugeno in the hydroponic sector with multiple inputs, ANFIS performs better in controlling the system; moreover, the accuracy of ANFIS in controlling the system is present | - |

## 3. Design of Smart Hydroponic Monitoring and Controlling System Using ANFIS

The system in this study is divided into two parts: hardware design and application design. Hardware design describes the whole system architecture and the way each

component communicates with each other. Application design explains the procedure on creating the ANFIS models and how the system works.

### 3.1. Proposed Hardware Design

Figure 1 depicts the design of the system, which is divided into two main components. The first component is the layer device, which includes a set of sensors, actuators, and a microcontroller. The second component is the layer database and web, which include a set of computation devices and an end-user interface.

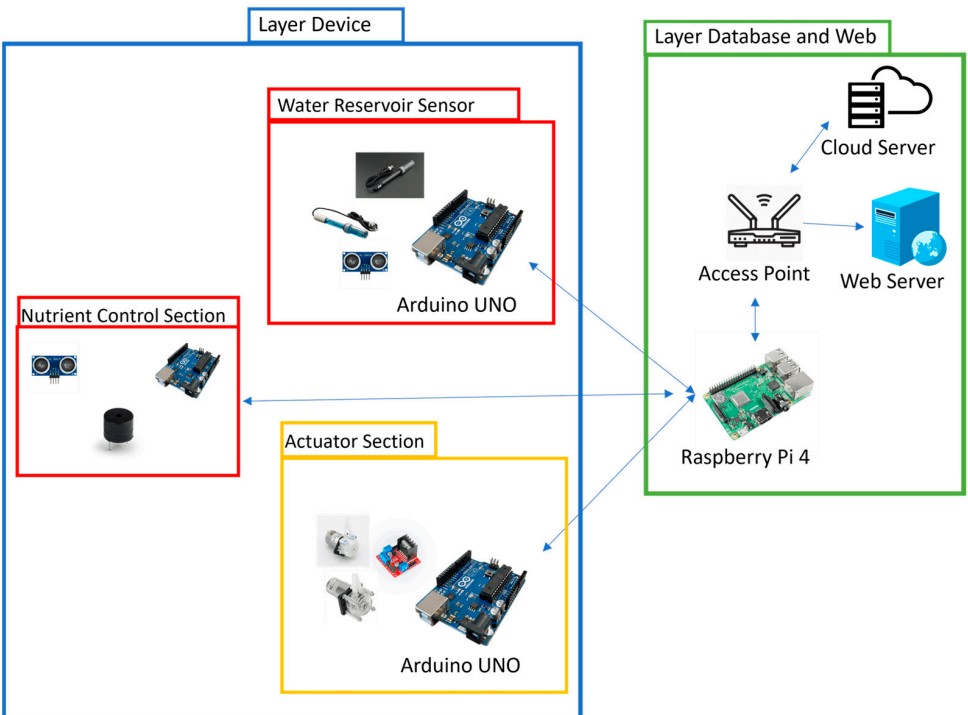

**Figure 1.** System architecture.

Layer device consists of three parts: a water reservoir sensor, a nutrient control section, and an actuator section. Figure 2 shows the details of the water reservoir sensor. In this part, Arduino Uno plays a very important role in controlling and receiving sensor data of pH and TDS. The flow of the sensor section starts from the ultrasonic sensor from Kuongshun Electronic HC-SR04 that is connected to Arduino Uno, which is used to measure the level of the water by calculating the distance between the top part of the water reservoir and the water in the reservoir itself.

Arduino is also connected to Dual H-Bridge L298N which is used to control the movement of the DC motor gearbox in accordance with the pulse width modulation (PWM) input from Arduino. Then, the TDS sensor from DFRobot DFR0300 and pH sensor from DFRobot SEN0161 V2 are attached to the DC motor gearbox, and based on the PWM input from Arduino, the TDS sensor and pH sensor will be lowered to the determined height to start measuring the level of TDS and pH of the water reservoir. After the measurement is performed, Arduino will receive the measured TDS and pH data.

Furthermore, there is also a nutrient control section in the device layer, the details as shown by Figure 3. The Arduino Uno plays a crucial role in ensuring that the pH and TDS chemicals are always readily available. Four ultrasonic sensors are connected to the Arduino and are employed to measure the levels of the chemical liquids for pH Up, pH Down, nutrient A, and nutrient B. In addition to the ultrasonic sensors, a Piezo buzzer is also connected to the Arduino, serving as a notification to the user when the chemical liquid level is running low. Hence, if the volume of any of the chemical liquids falls below

the predetermined level, the Piezo buzzer will produce sounds until the chemical liquid is returned to the specified level.

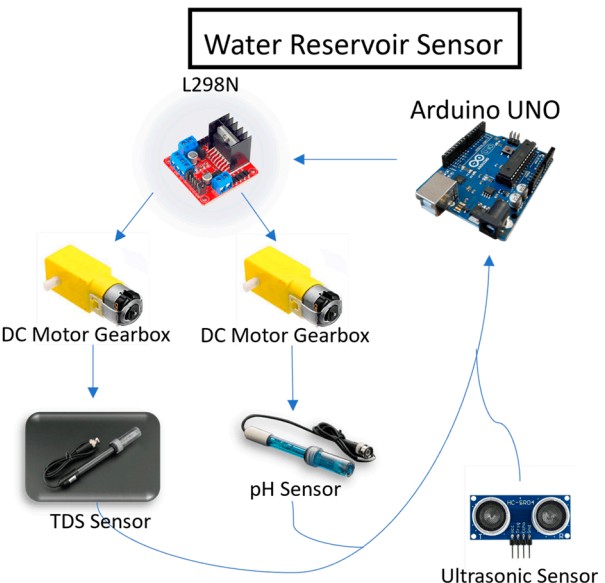

**Figure 2.** Water reservoir sensor details.

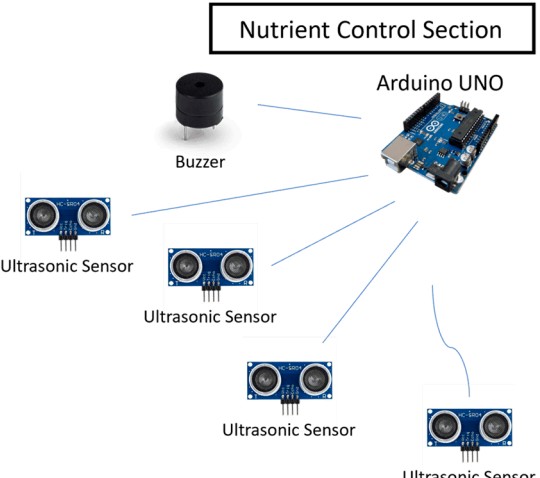

**Figure 3.** Nutrient control section details.

Arduino Uno also plays a critical role in the actuator section, serving as the control center for all existing actuators. As shown in Figure 4, there are two Dual H-Bridge L298N units connected to the Arduino, one to control the nutrient peristaltic pump and the other to control the pH peristaltic pump. The Dual H-Bridge L298N was chosen because it allows control of the actuators using PWM, enabling the precise control of the output of each peristaltic pump. Additionally, the Arduino is connected to a relay that is used to control three actuators: the water pump to add water to the reservoir, the water pump to reduce the water inside the reservoir, and the aerator to provide more oxygen into the system.

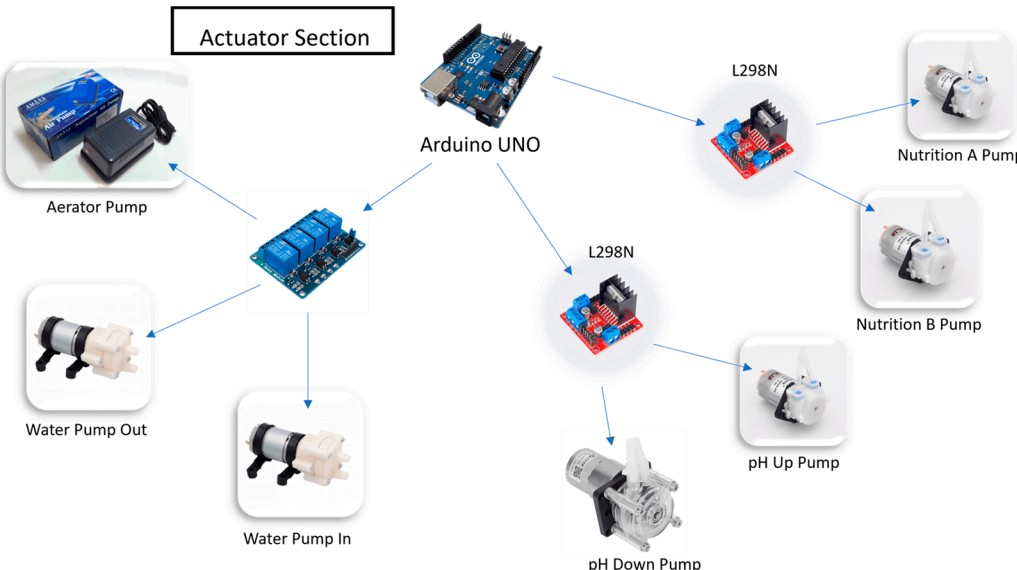

**Figure 4.** Actuator section details.

The Arduino on the water reservoir sensor section, nutrient control section, and actuator section are all connected to the Raspberry Pi 4 from the layer database and web, and this Raspberry acts as the brain of the entire system. The pH and TDS data on the Arduino sensor section are sent to Raspberry, then the pH and TDS data are processed by the fuzzy controller created by ANFIS, and the output is then sent to Arduino in the actuator section as shown in Figure 5. The output is a number in a form of seconds, which will determine how long each pump needs to be turned on in the actuator section. Raspberry will also send the output to the cloud database, then the web server will automatically detect that new data and reload the web component to show the latest data. This study also implements the internet of things, thus, the system can be controlled from anywhere. In order to accomplish everything described above, the Raspberry will work in multithreading by operating two applications at the same time. The first application is to control the system, while the other is to monitor the cloud database.

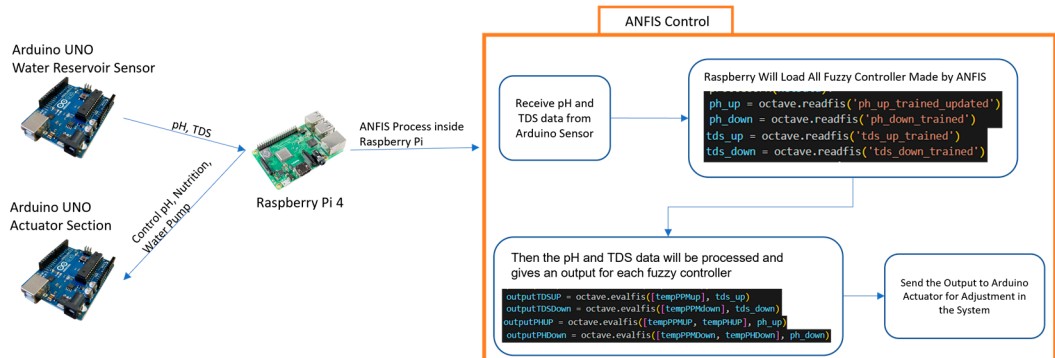

**Figure 5.** Raspberry Pi process details.

### 3.2. Proposed Application Design

ANFIS is a combination of fuzzy and neural network. The neural network needs a digital dataset to create the models and train it. The dataset needs to be accurate so it can produce a good ANFIS. However, a dataset for this study is not yet available. Therefore, as can be seen in Figure 6, the first step that needs to be taken is collecting data to be used as the study's dataset. The step in creating the dataset can be performed by conducting an experiment in adjusting the water's pH and nutrient level inside the reservoir based on the

pH and TDS data acquired by the sensors. The adjustments can be performed by pumping a pH up or down, by pumping a nutrient liquid, or by refilling the water in the reservoir. The time needed for the pump to be turned on to perform the adjustment will be recorded and stored to be used as a dataset. The pH and TDS sensors used for creating the dataset are manual tools as shown in Figure 7a,b. The accuracy of the pH and TDS meters has been tested and calibrated before being used.

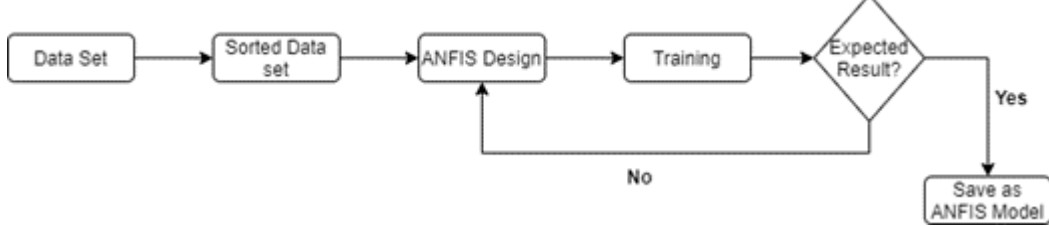

**Figure 6.** Data processing flow.

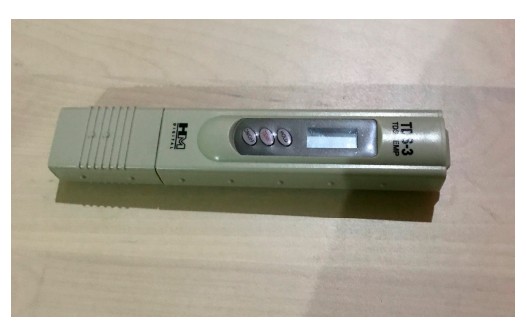

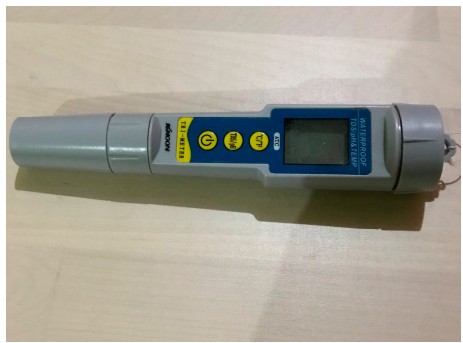

(**a**) Manual pH sensor     (**b**) Manual TDS sensor

**Figure 7.** Manual tools for creating the dataset.

The dataset is divided into four types, one for each ANFIS model, which results in four ANFIS models altogether. It needs to be divided into four models because the structure of ANFIS is made up of multi-input and single output. Therefore, each model will have an output that represents each actuator used to manipulate the level of pH and nutrients of the water inside the reservoir.

The first and the second datasets are the ones to lower or increase the level of pH of the water inside the reservoir. The first step in collecting the data is by filling the reservoir with 10 liters of water, then the pH is measured by using a pH meter. If the measured pH is less than normal, which is 5.5–6.5 [21], then the pH Up liquid will be pumped until it reaches the normal level. Meanwhile, if the measured pH is more than the normal level, then the pH Down liquid will be pumped until the pH level returns to normal. The duration of time that the pH liquid is turned on will be captured, as well as the pH and TDS data before and after manipulation.

The third and fourth datasets are the ones to lower or increase the level of nutrients dissolved in the water inside the reservoir. The steps to create the dataset uses the same steps as creating the pH dataset, which is by filling the reservoir with 10 L of water, then the dissolved nutrient will be measured. If the measured nutrient level is less than the normal threshold, which is 600–900 ppm [21], then the nutrient-boosting liquid will be pumped until it returns to the normal threshold. If it is more than its normal level, then the pump for disposing water from the reservoir and the pump for filling water into the reservoir will be turned on until the nutrient level returns to normal. The duration of time that each pump operates will be captured along with the pH and TDS data after the performed adjustment. Some examples of collected data can be seen in Table 3.

**Table 3.** Some examples of the dataset.

| EC | pH | Output Actuator pH Up (Second) | Output Actuator pH Down (Second) | Output Actuator TDS Up (Second) | Output Actuator TDS Down (Second) |
|---|---|---|---|---|---|
| 291 | 4.4 | 2.2 | 0 | 18 | 0 |
| 383 | 5.4 | 0.1 | 0 | 15 | 0 |
| 614 | 6.5 | 0 | 4.3 | 0 | 0 |
| 1090 | 6.1 | 0 | 0 | 0 | 240 |

The data cleansing process needs to be performed after the data have been collected to remove all duplicated and irrelevant data. The data can be duplicated because the data are created manually, which might happen because of double input. In addition, an example of irrelevant data is when there is a starting nutrient level and pH, but no nutrient level, and the pH is captured after adjusting the pH or TDS, thus causing the output result to be unusable as a dataset, which can happen because of wrong input. Microsoft Excel is used for cleansing the data; a simple conditional formatting is used to remove all of the duplicated and irrelevant data.

After the dataset is prepared, the design process of ANFIS can be started by determining the degree of membership function. Through trial and error, the most optimal membership functions are:

- For pH Up model, it is a generalized bell membership function, with 2 degrees of TDS membership function and 7 degrees of pH membership function, as shown in Figure 8;
- For pH Down model, it is a generalized bell membership function, with 2 degrees of membership function for TDS and 3 degrees of membership function for pH, as shown in Figure 9;
- For TDS Up model, it is a generalized bell membership function, with 5 degrees of membership function for TDS, as shown in Figure 10;
- For TDS Down model, it is a generalized bell membership function, with 5 degrees of membership function for TDS, as shown in Figure 11.

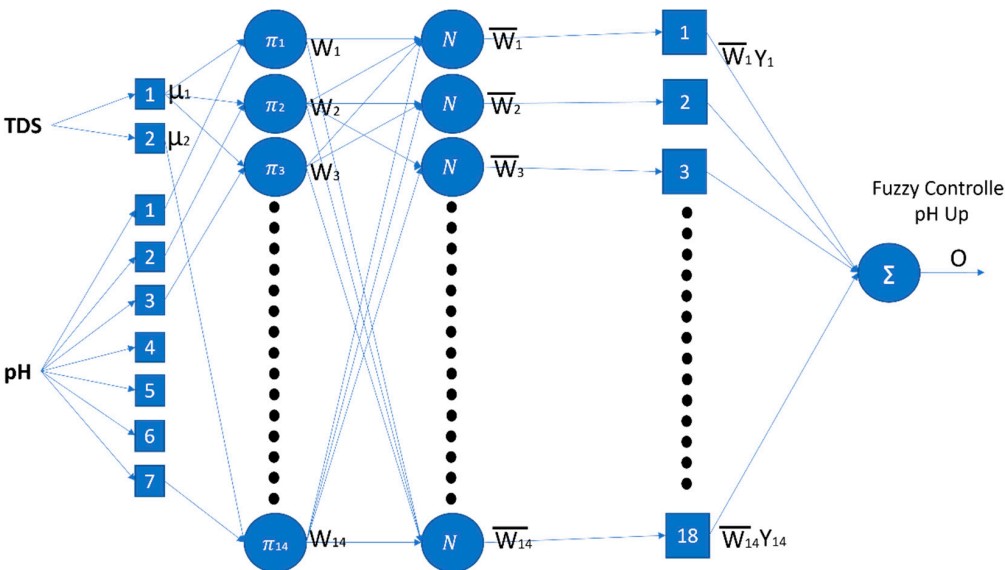

**Figure 8.** ANFIS pH Up structure.

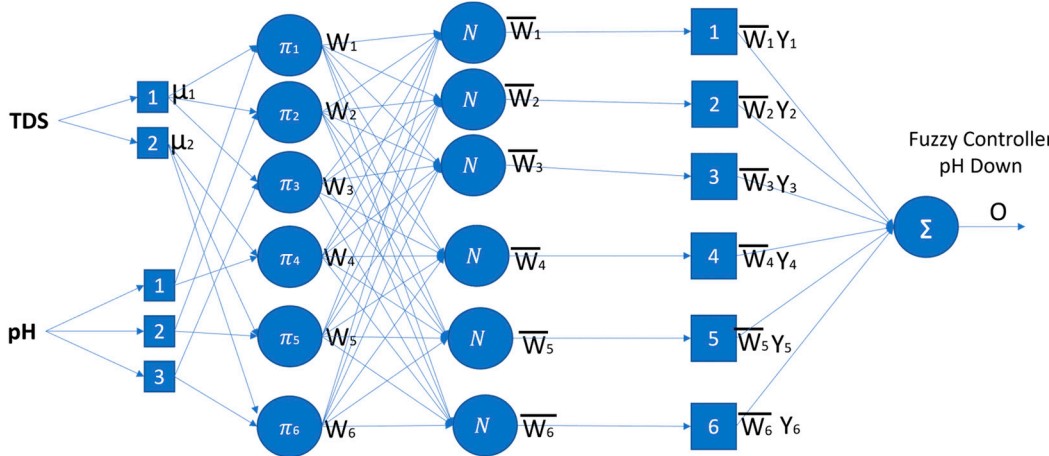

**Figure 9.** ANFIS pH Down structure.

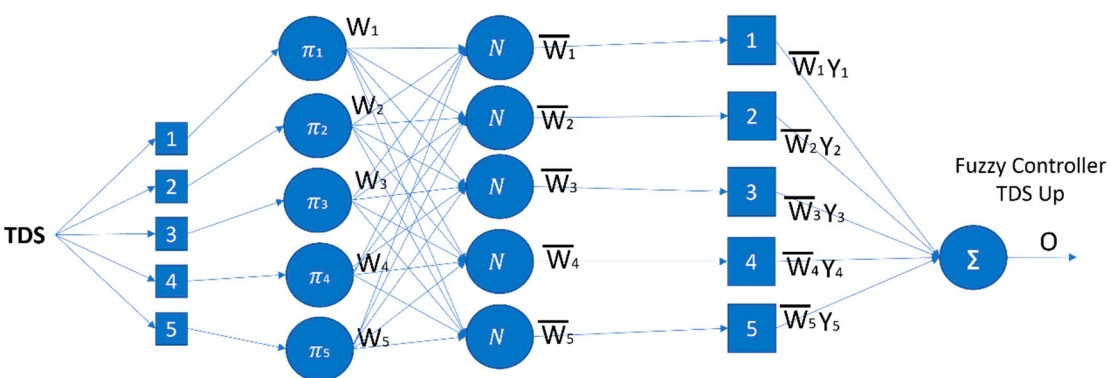

**Figure 10.** ANFIS TDS Up structure.

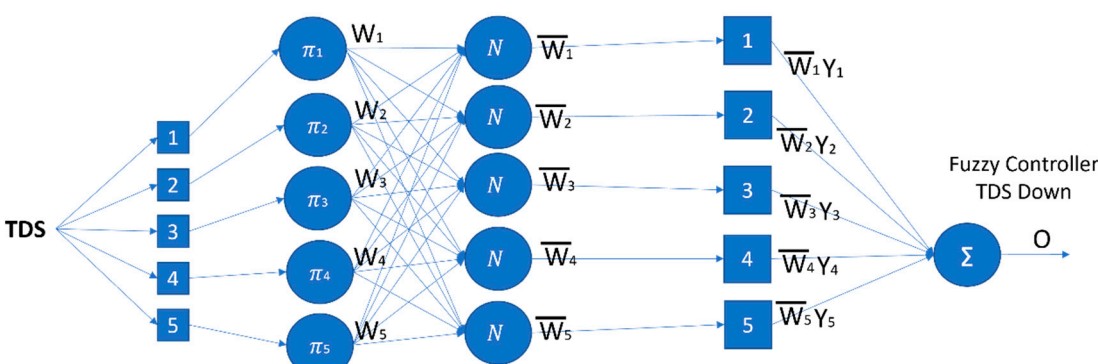

**Figure 11.** ANFIS TDS Down structure.

As for the Generalized bell membership function as follows:

$$f(x; a, b, c) = \frac{1}{1 + \left| \frac{x-c}{a} \right|^{2b}} \tag{1}$$

where $x$ is the input value, $c$ is the center of the bell-shaped curve, $a$ is the width of the curve, and $b$ is a shape parameter that controls the steepness of the curve. When $n$ is greater than 1, the curve becomes more steep, and when $n$ is less than 1, the curve becomes less steep. When $n$ approaches infinity, the curve becomes a step function.

The next phase after designing ANFIS is the training stage. In this stage, ANFIS will learn by itself based on the defined dataset. Then, an ANFIS model will be formed. An ANFIS model is a complete fuzzy controller that consists of membership functions of input-output and rule-base pairs.

Figure 12 shows the system workflow. When the system runs, there will be two processes that run at the same time. The first process is the automatic hydroponic adjustment process. When the time shows that it is measurement time, the initial step is measuring the level of the water inside the reservoir. If the level does not reach the threshold, which is 10 liters, then it needs to be redetermined whether the water level is below the threshold or not. If it is below the determined level, then the water pump to add water into the reservoir will turn on to add water until the water level is at 10 L. Meanwhile, if the water level is not below the threshold, then a validation needs to be carried out again to confirm whether the water is above the threshold. If it is true, then the water pump coming out of the reservoir will turn on to reduce the water in the reservoir until the water level is at 10 L. If the water level is currently at 10 L, then there will be a validation to ensure the status of the water pump that adding water into the reservoir is already off. If the status is still on, then a trigger will be sent to turn the pump off, but if it is off, then there will be a validation sent to ensure the status of the water pump that the function of pumping water out of the reservoir is switched off. If the status is still on, then a trigger is sent to turn off the pump, and if it is off, then the process will continue to measure pH and TDS. Next, an order will be given to Arduino to perform the pH measurement process and TDS measurement process. The captured pH and TDS data will be input into the ANFIS pH model. If the computational results of ANFIS pH is not 0, then the pH needs to be manipulated by adding a pH Up liquid or adding a pH Down liquid until it produces an output result of 0. The next process is TDS evaluation, where the captured TDS data will be computed on the Raspberry based on the created model. If the computation result is not 0, then the dissolved nutrient level will be manipulated by adding nutrient liquids A and B or adding pure water to the reservoir. The process will be performed repeatedly until the computation result is 0.

The second process is detecting the level of chemical liquid. There are four liquid chemicals used to manipulate the hydroponic system. Each liquid is stored inside a bottle with a height of 12 cm, and above the bottle, an ultrasonic sensor is embedded to measure the height of the liquid level. The first liquid to measure is pH Up. The ultrasonic pH Up sensor will read the liquid level. If the liquid level is below 3 cm, the Piezo buzzer will make a sound to notify the user that the pH Up liquid has run out. However, if the measurement results are above 3 cm, then the next liquid level measurement will continue until the last chemical liquid.

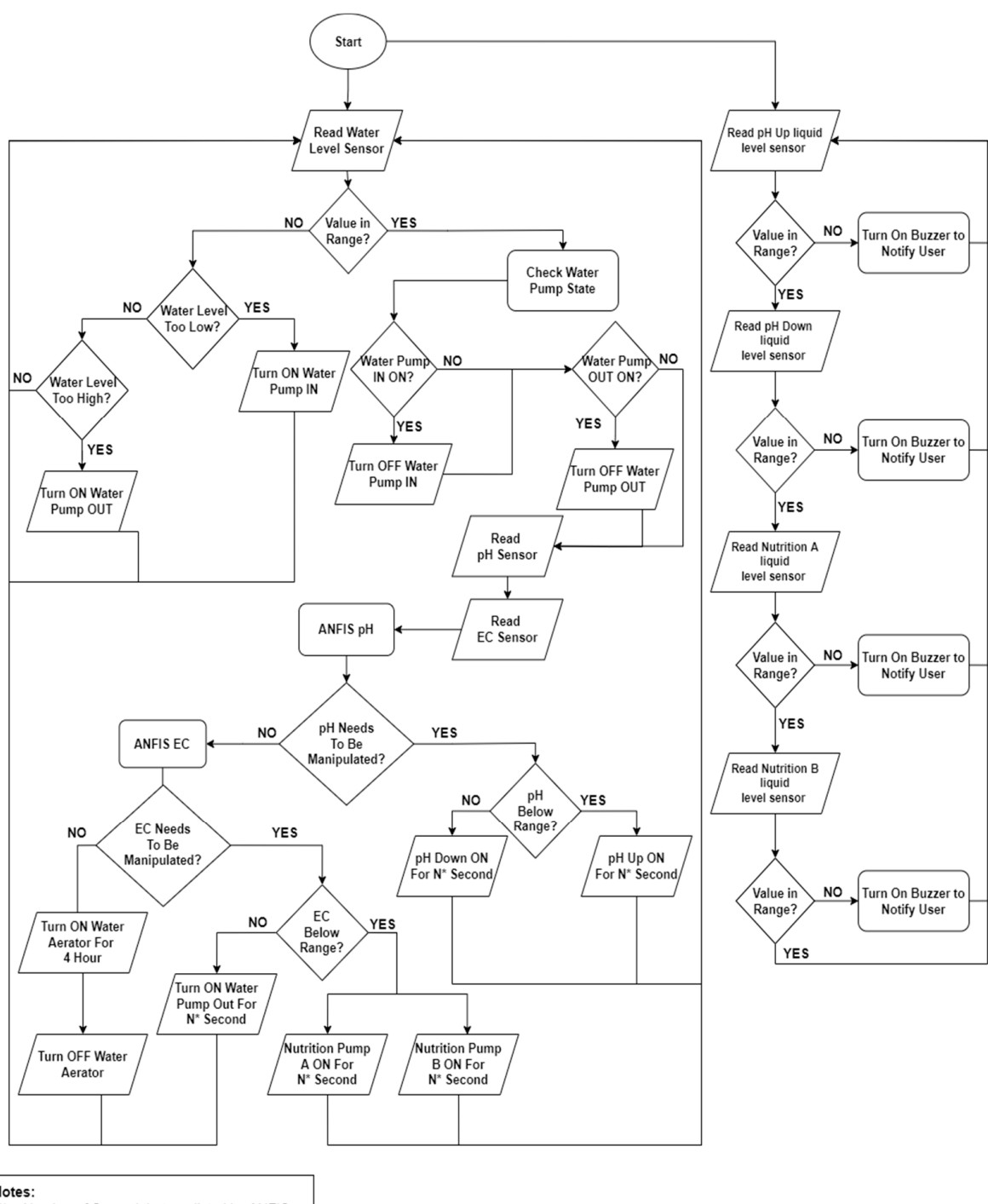

**Figure 12.** Flowchart system.

## 4. Results and Discussions

### 4.1. Device Installation & Implementation

The smart hydroponic system is made based on the design that has been described. Figure 13 is the final result of the system. The entire hydroponic system is placed indoors, so an LED glowing light is needed so that the plants can grow perfectly. Figure 14 shows Bok Choy that have been grown in a hydroponic system from the sowing process, to seedling and mature stages. It takes 30 days to grow until mature and after which, the Bok Choy have reached 52 cm in height.

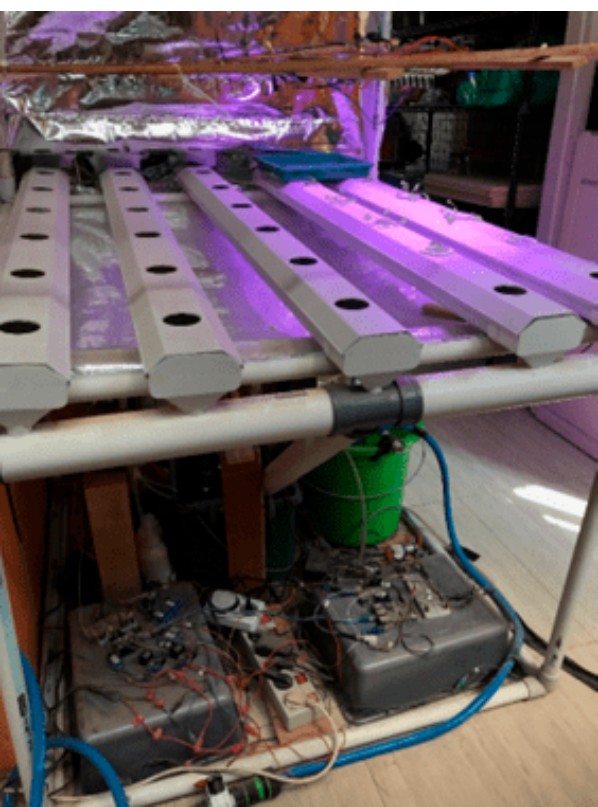

**Figure 13.** Final results of the smart hydroponic system.

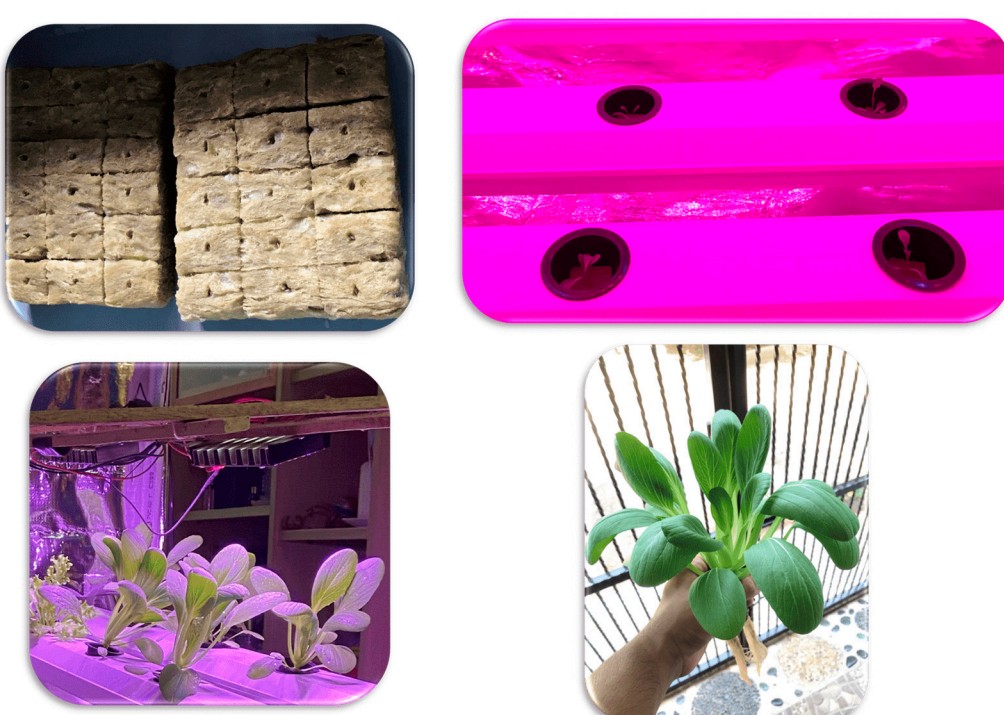

**Figure 14.** Bok Choy growing process.

Figure 15 shows the details of the parts used in the system. Each part has been numbered with the following details:

1.  pH Up peristaltic pump is used to increase the pH level in the reservoir;
2.  pH Down peristaltic pump is used to decrease the pH level in the reservoir;

3. Modified DC gearbox is used to lower and raise the pH and TDS sensors;
4. pH sensor is used to measure the pH level in the reservoir;
5. The TDS sensor is used to measure the TDS level in the reservoir;
6. Ultrasonic sensor is used to measure the level of water inside the reservoir;
7. The hydroponic system water pump is used to pump water from the reservoir to the NFT hydroponic system;
8. pH up, pH down, nutrient A, and nutrient B chemicals are used to manipulate pH and TDS levels in the reservoir;
9. TDS Up peristaltic pump is used to increase the nutrient or TDS levels in the reservoir;
10. First water pump is used to add water to the main reservoir from the temporary water reservoir;
11. Temporary water storage is used to store water before it is put into the system;
12. Second water out is a pump used to reduce the water in the reservoir;
13. Relays are used as on or off switches for all of the water pumps;
14. Dual H-Bridge L298N is used to activate the DC gearbox;
15. Arduino Uno is used as a microcontroller for pH, TDS, and ultrasonic sensors;
16. Raspberry Pi 4 acts as the brain of the smart hydroponic system;
17. Dual H-Bridge L298N is used to turn on the TDS Up peristaltic pump;
18. Dual H-Bridge L298N is used to turn on the pH Up and pH Down peristaltic pumps;
19. Arduino Uno is used as a microcontroller for all actuators.

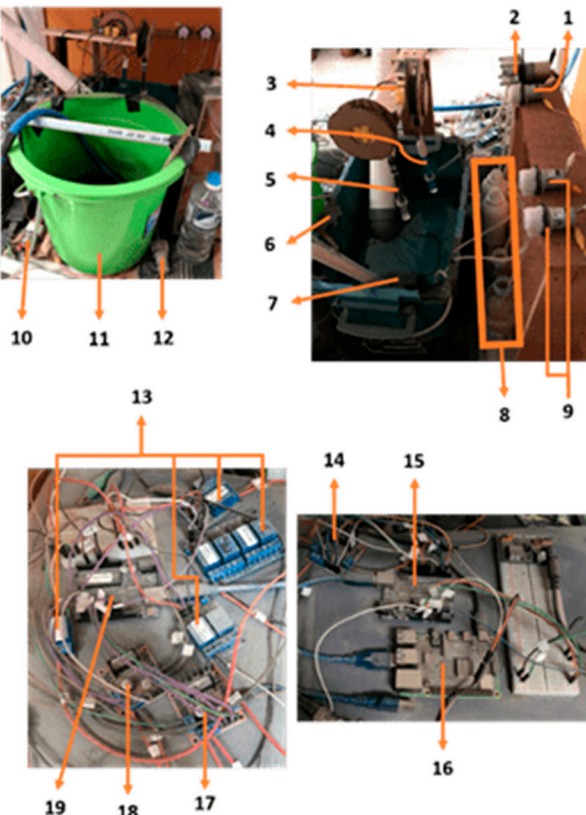

**Figure 15.** Details of the smart hydroponic system.

### 4.2. System Evaluation

In order to ensure that the system has been designed correctly, several evaluations need to be conducted. The first evaluation is calculating the pH and TDS sensors error rate. The error rate can be calculated by comparing the results of system sensor measurements

with the manual sensors used in data collection. The calculation of error rate can be seen as follows:

$$\%Error = \left( \frac{Original\ Value - Measured\ Value}{Original\ Value} \right) \times 100\% \qquad (2)$$

where *Original Value* is the manual sensors, as can be seen in Figure 7, and *Measured Value* is the sensor that is used in the system, as can be seen in Figure 15. Tables 4 and 5 show the error rate of each sensor. The "pH Data Collection" and "TDS Data Collection" columns are the data that have been captured by the manual sensors. The "pH System" and "TDS system" columns are the data that have been captured by the system sensors.

**Table 4.** Error rate pH sensor.

| No | pH Data Collection | pH System | Error Rate |
|----|--------------------|-----------|------------|
| 1 | 3.32 | 3.26 | 1.81 |
| 2 | 3.68 | 3.65 | 0.82 |
| 3 | 3.87 | 3.95 | −2.07 |
| 4 | 4.35 | 4.28 | 1.61 |
| 5 | 4.98 | 4.9 | 1.61 |
| 6 | 6.78 | 6.94 | −2.36 |
| 7 | 7.12 | 7.38 | −3.65 |
| 8 | 7.8 | 7.77 | 0.38 |
| 9 | 8.13 | 8.37 | −2.95 |
| 10 | 9.71 | 10.4 | −7.11 |
| Average Error | | | −1.19 |

**Table 5.** Error rate TDS sensor.

| No | TDS Data Collection | TDS System | Error Rate |
|----|---------------------|------------|------------|
| 1 | 295 | 291 | 1.36 |
| 2 | 374 | 399 | −6.68 |
| 3 | 452 | 445 | 1.55 |
| 4 | 534 | 552 | −3.37 |
| 5 | 592 | 583 | 1.52 |
| 6 | 595 | 598 | −0.50 |
| 7 | 941 | 952 | −1.17 |
| 8 | 1060 | 1090 | −2.83 |
| 9 | 1210 | 1213 | −0.25 |
| 10 | 1410 | 1399 | 0.78 |
| Average Error | | | −0.96 |

The average error rate for pH sensors is −1.19% and the average error rate for TDS sensors is −0.96%. Both sensors have very small error rates which prove that the pH and TDS system sensors are already accurate. After they have been proven to be successful, then a test on each created model should be performed to prove that those models are working properly.

As for the ANFIS pH model, the first step that should be taken is manipulating the level of pH inside the reservoir, whether by increasing or decreasing the pH level. Then, the system will try to normalize the pH level inside the reservoir in accordance with the manipulation.

Table 6 shows the response of the ANFIS pH model during the testing. "Starting Data" is the pH and TDS levels in the reservoir after the manipulation step by increasing or decreasing the pH level. The "Starting Data" are the input for the ANFIS pH model. Then, ANFIS pH model will provide output data in the form of the duration to turn on the pH Up or pH down pump in seconds. After that, the pH and TDS measurements will be repeated. The result of the measurement after increasing or decreasing the pH level based

on the ANFIS output is captured. It can be seen on Table 4 that the ANFIS pH model has succeeded in returning the manipulated pH levels to the normal range.

**Table 6.** Testing results of ANFIS pH model.

| No | Starting Data | | Action | Duration (Second) | Result Data | | Expected Result |
|---|---|---|---|---|---|---|---|
| | pH | TDS | | | pH | TDS | |
| 1 | 7.28 | 631 | pH Down ON | 4.21 | 6.21 | 620 | Correct |
| 2 | 6.6 | 310 | pH Down ON | 0.6 | 6.35 | 310 | Correct |
| 3 | 3.43 | 783 | pH Up On | 15 | 6.38 | 767 | Correct |
| 4 | 3.26 | 1105 | pH Up On | 27.3 | 5.92 | 1029 | Correct |

Table 7 shows the response of the ANFIS TDS model during the testing. "Starting Data" is the pH and TDS levels in the reservoir after the manipulation step by increasing or decreasing the nutrient level. The "Starting Data" are the inputs for the ANFIS TDS model. Then ANFIS TDS model will provide output data in the form of the duration to turn on the nutrients A and B pump or adding fresh water to the reservoir in seconds. After that the pH and TDS measurements will be repeated. The result of the measurement after increasing or decreasing the TDS level based on the ANFIS output is captured. It can be seen on Table 6 that the ANFIS TDS model has succeeded in returning the manipulated TDS levels to the normal range.

**Table 7.** Testing results of ANFIS TDS model.

| No | Starting Data | | Action | Duration (Second) | Result Data | | Expected Result |
|---|---|---|---|---|---|---|---|
| | pH | TDS | | | pH | TDS | |
| 1 | 6.35 | 310 | TDS Up ON | 18.1 | 6.3 | 650 | Correct |
| 2 | 6.35 | 552 | TDS Up ON | 10.9 | 6.3 | 829 | Correct |
| 3 | 6.46 | 1228 | TDS Down ON | 317 | 7.3 | 660 | Correct |
| 4 | 6.1 | 1032 | TDS Down ON | 224 | 6.86 | 702 | Correct |

*4.3. ANFIS Evaluation*

The accuracy of the ANFIS model needs to be compared to show that the model provides a better accuracy. In this research, the ANFIS model will be compared with fuzzy Sugeno. The data collection process, such as when creating a dataset, needs to be conducted again, so that this data will become a benchmark in measuring the level of accuracy of the two models. The data is labeled as "Observed Data". The benchmark data that have been collected can be seen in Table 8.

**Table 8.** Observed data.

| Starting Point | | Output Actuator (Second) | | | | Result | |
|---|---|---|---|---|---|---|---|
| pH | TDS | pH Down | pH Up | TDS Up | TDS Down | pH | TDS |
| 2.87 | 475 | 0 | 20 | 0 | 0 | 6.15 | 690 |
| 3.34 | 1210 | 0 | 25 | 0 | 0 | 6.02 | 1230 |
| 3.72 | 535 | 0 | 6 | 0 | 0 | 5.8 | 508 |
| 2.93 | 1320 | 0 | 40 | 0 | 0 | 5.67 | 1010 |
| 4.15 | 1130 | 0 | 4.25 | 0 | 0 | 5.5 | 1140 |
| 4.63 | 829 | 0 | 2.1 | 0 | 0 | 5.61 | 837 |
| 8.12 | 803 | 8.9 | 0 | 0 | 0 | 6.46 | 807 |
| 8.6 | 905 | 9 | 0 | 0 | 0 | 5.58 | 952 |
| 6.76 | 1080 | 2 | 0 | 0 | 0 | 6.48 | 1070 |
| 7.21 | 425 | 3.8 | 0 | 0 | 0 | 6.38 | 417 |
| 10.18 | 924 | 14.5 | 0 | 0 | 0 | 6.46 | 911 |
| 7.95 | 1182 | 7.9 | 0 | 0 | 0 | 6.48 | 1450 |
| 6.49 | 413 | 0 | 0 | 14 | 0 | 6.5 | 685 |

**Table 8.** *Cont.*

| Starting Point | | Output Actuator (Second) | | | | Result | |
| pH | TDS | pH Down | pH Up | TDS Up | TDS Down | pH | TDS |
|---|---|---|---|---|---|---|---|
| 6.07 | 296 | 0 | 0 | 18 | 0 | 6.08 | 710 |
| 6.38 | 320 | 0 | 0 | 16.5 | 0 | 6.41 | 693 |
| 6.41 | 548 | 0 | 0 | 11.3 | 0 | 6.42 | 794 |
| 6.28 | 825 | 0 | 0 | 0 | 0 | 6.28 | 825 |
| 6.48 | 1450 | 0 | 0 | 0 | 552 | 6.96 | 610 |
| 6.5 | 997 | 0 | 0 | 0 | 192 | 6.61 | 725 |
| 5.62 | 1020 | 0 | 0 | 0 | 240 | 5.82 | 670 |
| 6.44 | 1210 | 0 | 0 | 0 | 320 | 6.87 | 690 |
| 6.39 | 721 | 0 | 0 | 0 | 0 | 6.39 | 721 |

The next step is creating the fuzzy Sugeno. In fuzzy Sugeno, four fuzzy controllers will be made, fuzzy controller pH Up, fuzzy controller pH Down, fuzzy controller TDS UP, and fuzzy controller TDS Down. The first process is to define all degrees of the membership function for each fuzzy controller. According to research by Herman and Surantha [13], the degree of membership function is as shown in Figure 16, and pH can be categorized as follows:

- Very Low (0–3.3);
- Low (3–5.5);
- Normal (5.25–6.5);
- Moderately High (6.25–7.75);
- High (7.9–9.25);
- Very High (9–14).

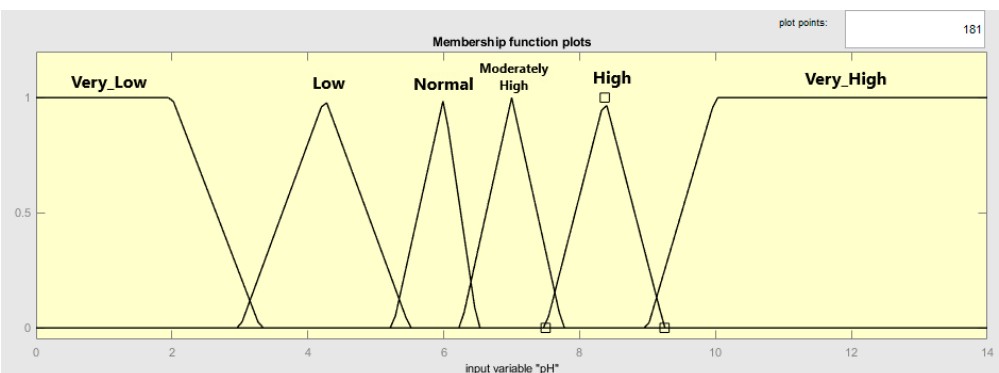

**Figure 16.** Membership function of fuzzy controller pH.

As for the degree of TDS membership function, it can be seen in Figure 17, where TDS can be categorized as follows:

- Very Low (0–330);
- Low (320–495);
- Moderately Low (485–600);
- Normal (590–910);
- Moderately High (900–1000);
- High (990–1400);
- Very High (1390–1600).

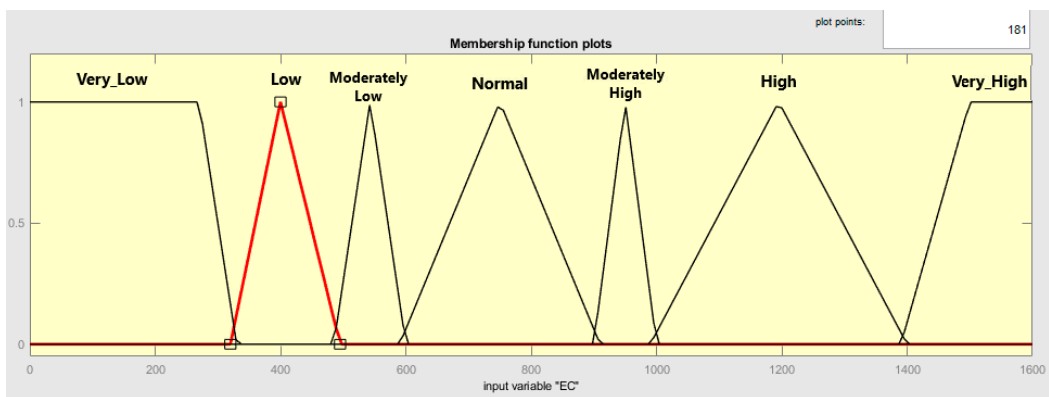

**Figure 17.** Membership function fuzzy of controller pH.

Then, the rules for each condition needs to be defined. Tables 9 and 10 are knowledge bases for pH and as a guidance for each IF–Then statement. As for the output of fuzzy, which is in the form of how long the pH Up and pH Down need to turned on can be seen in Figure 18.

**Table 9.** Knowledge base for pH Up.

| EC \ pH | Very Low | Low | Normal | High | Very High |
|---|---|---|---|---|---|
| Normal | Idle | Idle | Idle | Idle | Idle |
| Low | Moderate | Fast | Fast | Moderate | Long |
| Very Low | Moderate | Moderate | Moderate | Long | Long |

**Table 10.** Knowledge base for pH Down.

| EC \ pH | Very Low | Low | Normal | High | Very High |
|---|---|---|---|---|---|
| Normal | Idle | Idle | Idle | Idle | Idle |
| Moderately High | Fast | Fast | Moderate | Moderate | Moderate |
| High | Moderate | Moderate | Moderate | Moderate | Long |
| Very High | Moderate | Moderate | Long | Long | Long |

**Output Level pH Up**

| Output Parameter | Duration (Second) |
|---|---|
| Fast | 0–11 |
| Moderate | 10–21 |
| Long | 20–31 |
| Very Long | 30–40 |

**Output Level pH Up**

| Output Parameter | Duration (Second) |
|---|---|
| Fast | 0–6 |
| Moderate | 5–11 |
| Long | 10–20 |

**Figure 18.** Output level of pH.

Once all fuzzy controllers have been made, the last evaluation can then be performed by giving the fuzzy Sugeno and ANFIS the same starting data or input data as the benchmark data, then the output results can be compared with the output based on Table 8. The details can be seen in Table 11.

**Table 11.** Result evaluation.

| Starting State | | Which Fuzzy Controller Activated | Output Actuator (Second) | | Target |
| pH | TDS | | Fuzzy Sugeno [13] | Fuzzy ANFIS | |
|---|---|---|---|---|---|
| 2.87 | 475 | pH Up | 15.4 | 19.6 | 20 |
| 3.34 | 1210 | pH Up | 25.4 | 25.5 | 25 |
| 3.72 | 535 | pH Up | 5.36 | 5.49 | 6 |
| 2.93 | 1320 | pH Up | 38.3 | 40.7 | 40 |
| 4.15 | 1130 | pH Up | 5.37 | 4.27 | 4.25 |
| 4.63 | 829 | pH Up | 9.08 | 3.42 | 2.1 |
| 8.12 | 803 | pH Down | 8 | 8.27 | 8.9 |
| 6.76 | 1080 | pH Down | 6.64 | 2.42 | 2 |
| 8.6 | 905 | pH Down | 8 | 9.62 | 9 |
| 7.21 | 425 | pH Down | 3 | 3.69 | 3.8 |
| 10.18 | 924 | pH Down | 13.1 | 11.1 | 14.5 |
| 7.95 | 1182 | pH Down | 10.4 | 9.04 | 7.9 |
| 6.49 | 413 | TDS Up | 16 | 14.6 | 14 |
| 6.07 | 296 | TDS Up | 19.4 | 17.7 | 18 |
| 6.38 | 320 | TSD Up | 19.2 | 16.8 | 16.5 |
| 6.41 | 548 | TSD Up | 7 | 10.9 | 11.3 |
| 6.28 | 825 | TDS Up | 0.1 | 0.06 | 0 |
| 6.48 | 1450 | TDS Down | 609 | 545 | 552 |
| 6.5 | 997 | TDS Down | 211 | 224 | 192 |
| 5.62 | 1020 | TDS Down | 335 | 224 | 240 |
| 6.44 | 1210 | TDS Down | 335 | 305 | 320 |
| 6.39 | 721 | TDS Down | 0 | 0 | 0 |

The root mean square error (RMSE) needs to be calculated based on the results from each output in Table 11 in order to determine the difference between the output predicted by the model and the observed data. Afterwards, a box plot should be created to visually depict the distribution of the data for an easier analysis. The RMSE box plots of all fuzzy controller models as shown in Figures 19–22

$$RMSE = \sqrt{\left(forecast\ values - observed\ values\right)^2} \tag{3}$$

where *forecast values* is the output that is predicted by fuzzy Sugeno or fuzzy ANFIS, and *observed values* is the data that have been captured manually and used for the dataset. Referring to the RMSE box plots of all fuzzy controller models as shown in Figures 19–22, it can be seen that the ANFIS fuzzy controller produces a smaller range and error rate than the Sugeno fuzzy controller. Furthermore, the percentage of improvement by using ANFIS instead of fuzzy Sugeno can be calculated by averaging all of the RMSE. The results can be seen in Table 12.

**Table 12.** ANFIS percentage of improvement.

| Model | Average RMSE | | Percentage of Improvement |
| | Fuzzy Sugeno [13] | ANFIS | |
|---|---|---|---|
| pH Up | 2.5733333 | 0.575 | 78% |
| pH Down | 1.8733333 | 1.053333333 | 44% |
| TDS Up | 2.1 | 0.332 | 84% |
| TDS Down | 37.2 | 14 | 62% |
| Average Percentage | | | 67% |

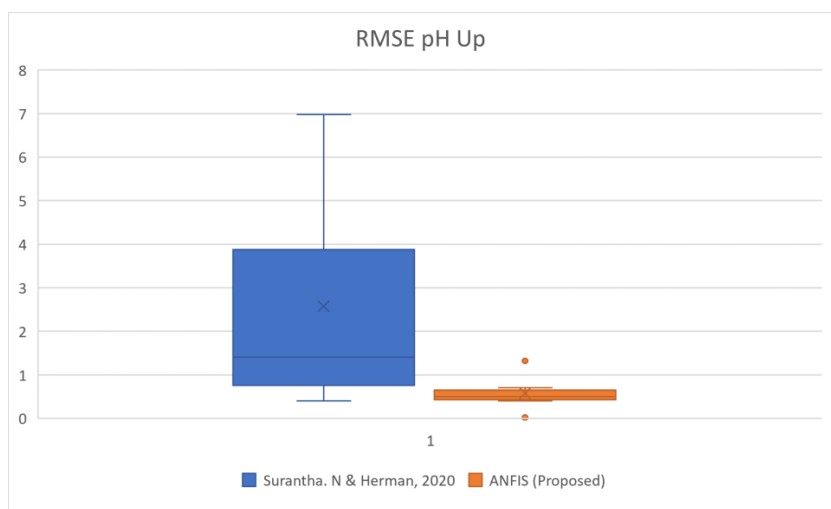

**Figure 19.** Box plot RMSE pH Up.

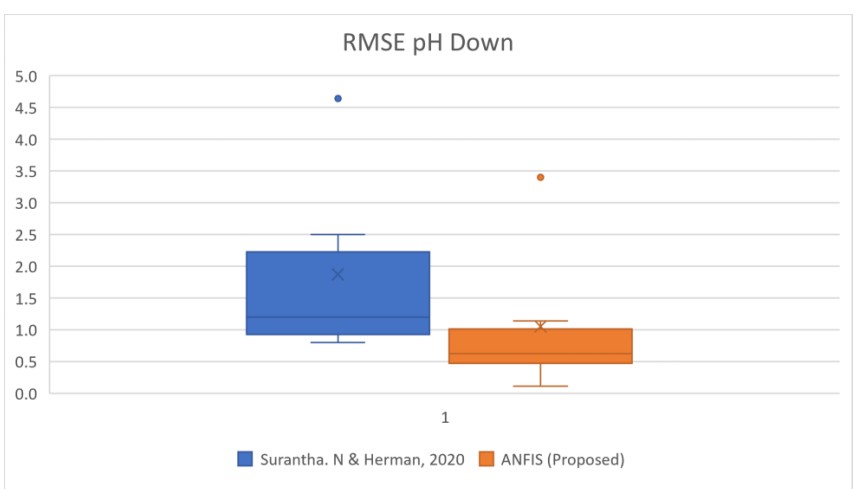

**Figure 20.** Box plot RMSE pH Down.

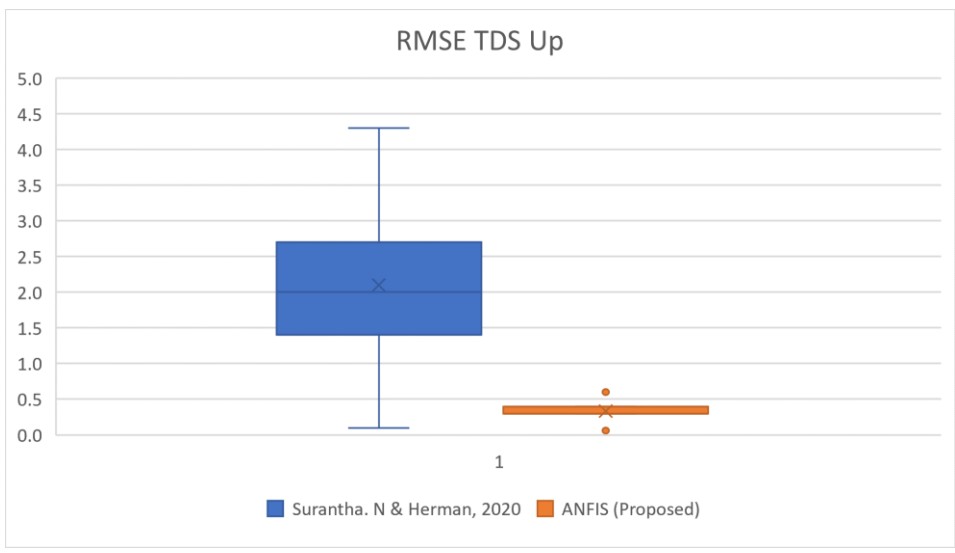

**Figure 21.** Box plot RMSE TDS Up.

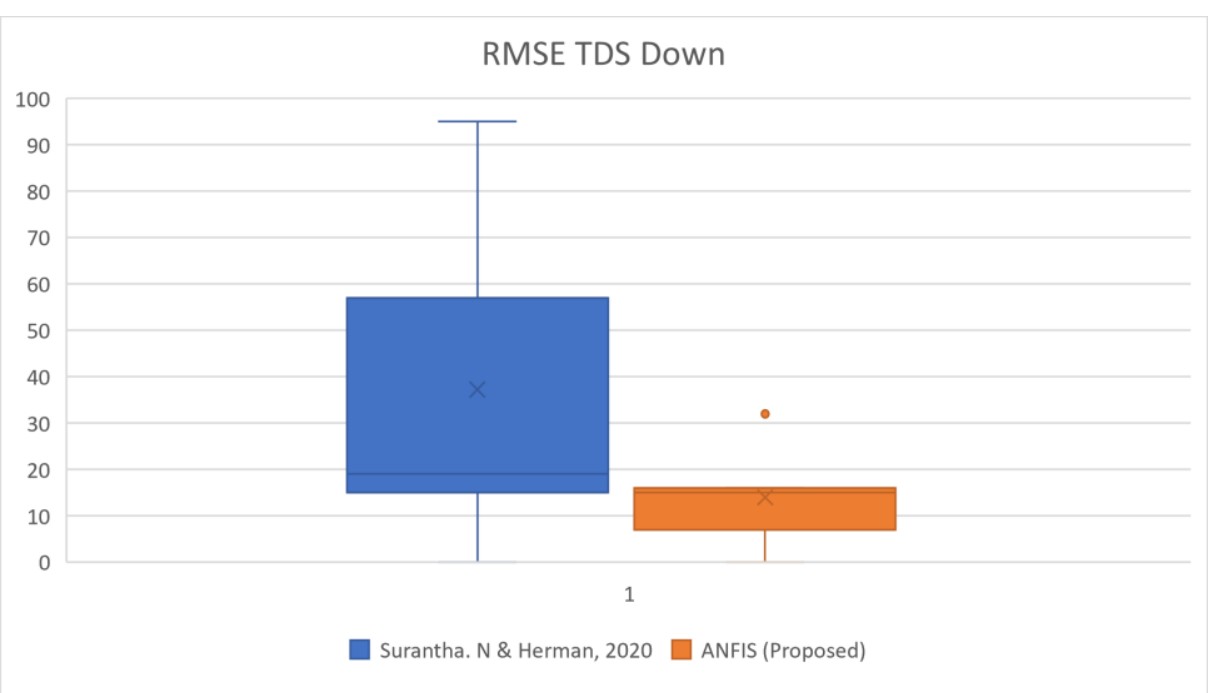

**Figure 22.** Box plot RMSE TDS Down.

Thus, it can be concluded that the output results from the ANFIS fuzzy controller model are more stable and 67% more accurate than the Sugeno fuzzy controller. Therefore, it can also be said that the implementation of the ANFIS method in making fuzzy controllers for controlling smart hydroponic systems provides a better predictive result compared with the Sugeno method.

### 4.4. Smart Hydroponic Web Application

Figure 23 illustrates the dashboard smart hydroponic web app page. The IoT features can be obtained by using this web application. There are six pieces of information on this page:

1. Plant Type, which provides information about the crops that are being planted in the system;
2. Planted, which provides information about when the plant is planted;
3. Global Status, which informs the current plant growth conditions based on TDS and pH levels;
4. Growing Week, which provides information about how many weeks the plants have been in the system since they were planted;
5. Tank Condition, which provides an updated information on PPM, pH, and water level status in the reservoir. PPM and pH are shown with a gauge meter with the PPM range of 1–1250, and for pH range of 1–14;
6. Environment Condition, which informs the status of the system environment. Here, the user can turn on or off the system from anywhere by pressing the "Main Power" button. In addition, the user can also turn on or turn off the "Grow Light" lamp. Lastly, the information of the weather forecast chart for the next 7 days can be accessed.

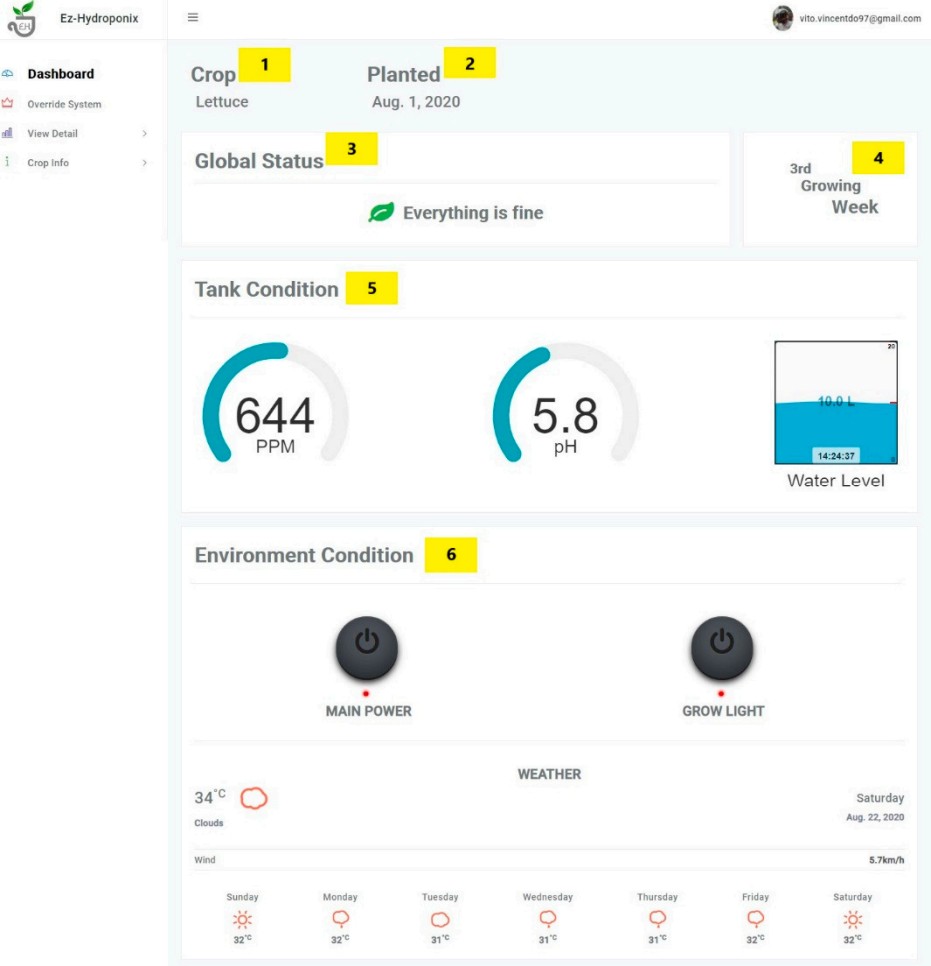

**Figure 23.** Details of the smart hydroponic system.

This system can also be manually operated by the users through the "Override System" menu, as can be seen in Figure 24. Users only need to turn on the feature, then fill in the desired pH and PPM values.

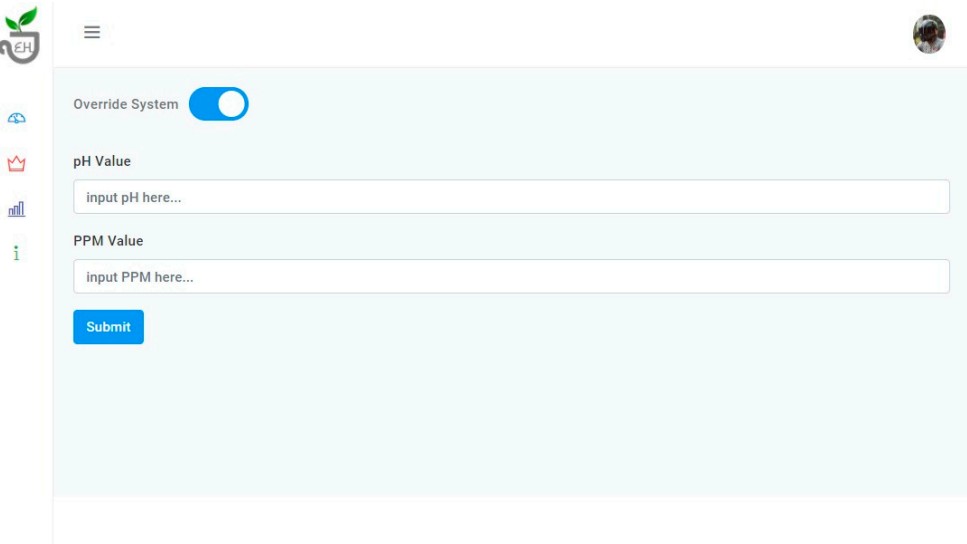

**Figure 24.** Override menu in web application.

## 5. Conclusions

Developing automatic hydroponic monitoring and controlling systems can help both traditional farmers and hydroponic farmers. As for the traditional farmers, the existence of this system can be a consideration for them to change their plantation medium to make it more efficient. Meanwhile, for the hydroponic farmers, this system can help them in reducing the routines that they need to perform, which involves checking the levels of nutrients, pH, and water levels.

In developing a smart hydroponic system, a system that can perform accurate adjustments on many actuators is needed. This kind of system can be developed by using the fuzzy logic method. However, this method requires the knowledge of an expert in defining membership functions and rules. Therefore, a different fuzzy method that combines artificial intelligence is needed so that the system can perform independent learning based on the existing data and eliminate the dependency of an expert's knowledge. The method that fulfils this requirement is ANFIS, which is a method that combines neural networks and fuzzy logic. This method can create fuzzy controllers independently from the given dataset. Thus, to make ANFIS produce maximum performance and accuracy, a good dataset is needed. This study also proves that the implementation of the ANFIS method is superior in accuracy than the traditional fuzzy method, which is Sugeno fuzzy, by 67%.

As for future work, the system can be improved by incorporating GoAT ranking sensors to optimize the system flow [12]. Additionally, by implementing a multi-adaptive neuro fuzzy inference system (MANFIS), the system can be made more efficient, as only one fuzzy controller model will be required instead of four [22]. Furthermore, the system can be expanded to a larger scale and integrated with edge-fog-cloud computing environments to achieve near real-time data processing, reduce latency, and increase system efficiency [4]. Finally, conducting a study to compare the growth and yields of plants grown in a NFT system, such as broccoli and spinach, with traditional planting methods, can provide insights into the advantages of this method compared to traditional methods.

**Author Contributions:** Conceptualization, V.V. and N.S.; methodology, N.S.; software, V.V.; validation, V.V.; formal analysis, N.S.; investigation, V.V.; resources, N.S.; data curation, V.V.; writing—original draft preparation, V.V. and N.S.; writing—review and editing, V.V.; visualization, V.V.; supervision, N.S.; project administration, V.V.; funding acquisition, N.S. All authors have read and agreed to the published version of the manuscript.

**Funding:** This research publication is fully supported by Bina Nusantara University.

**Data Availability Statement:** The data presented in this study are available on request from the corresponding author. The data are not publicly available due to privacy purposes.

**Conflicts of Interest:** The authors declare no conflict of interest.

## Nomenclature

| | |
|---|---|
| MHz | Unit of electromagnetic wave frequency, megahertz |
| GHz | Unit of electromagnetic wave frequency, gigahertz |
| Ppm | Concentration of solute in very dilute solutions, parts per million |
| Cm | length, centimeter |

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
