# Peer review of "Nutrient Film Technique-Based Hydroponic Monitoring and Controlling System Using ANFIS"

_electronics, doi:10.3390/electronics12061446_

Round 1

Reviewer 1 Report

This is an interesting research work. The following suggestions target to improve this paper contributions:

a) Abstract: 

> the Adaptive Neuro Fuzzy Inference System (ANFIS) must appears in this way, because is the first time mentioned.

> The sentence " ...using ANFIS gave more precise performance of prediction accuracy than the fuzzy controller made by using Sugeno" could be probably considered as " ...using ANFIS gave more precise performance of prediction accuracy than the Sugeno fuzzy controller." (?)

b) Section 3.1 - Refering to the  Figure 1. System Architecture.

> The actual contemporaneous architectures considered EDGE-FOG-Cloud as the more correct environment to support this study case problem. 

> Therefore, it is strong suggest in the Conclusions section a paragraph of future work considering this architecture proposal improvement, This is special interesting for local storage, processing, and latency issues. 

c) Section 2, 3, 4 and 5 - The word DATA is employed without comments that these data is DIGITAL DATA which requires observation to digital data storage, processing and latency to send to be finnaly storage in the cloud.

> The agriculture digital data is differenciated, because it is not process comes from digitation or digitalization. It is a natural DIGITAL DATA, which must be mentioned.  

> This is the "new scenario" in several applications. Therefore, it is suggested to provide references about "survey of near real time", which are approprieted to the agricultural sector field.  A fast search for "survey of near real time" will bring research work to be mentioned and provide more quality to the present contribution.

d) Finnaly a sensor ranking approach also could be suggested as a future work. Again, a research search for "sensor ranking" will indicate the importance to exist inside the future research indicarion

Author Response

Thank you for your valuable comments and suggestions. Please check our reply in the attachment file.

Reviewer 2 Report

The nutrient level is an important parameter for the NFT based hydroponic system, this manuscript created the monitoring and control system based on the IoT and ANFIS control model. The topic and results in this paper are interesting, and I just had a few questions below:

1.     The section 2 related works: the IoT and ANFIS were used for the monitoring and system for the hydroponic system, it is necessary to the introduce more literatures to describe the research status and the lack of the research to show the novelty of this study.

2.     Line 74 Please confirms the parameter about the PPM.

3.     The Figure 2, Figure 3 and Figure 4 were the parts of the Figure 1, and the Figure 3 is about the nutrient level control, The title of the “Chemical sensor” maybe cause the misunderstand about the chemical sensors.

4.     All the sensors of the system need be described, such as the type, the manufacturer.

5.     The parameter EC was described in the section 3.2, but the IoT system in section 3.1 and the section 4 was about the TDS, please confirm these two parameters.

6.     Table 7: the results about the control system provided the working time of the output actuator, it is necessary to show the target and actual values about the control for pH, the TDS.

7.     The system was used for the hydroponic system, so the results about the plant growth was lacked

8.     There are many researches about the IoT system and the ANFIS control model for the NFT based hydroponic system. The novelty of this manuscript was poor.

Author Response

(The authors gave the same response as above.)

Reviewer 3 Report

The background information provided in the introduction can be appropriately increased. At present, all the cited references are relevant to the research, but there are few references in general; The research is designed appropriately. The experimental steps and principles are described in the paper, but you can simply explain the meaning of the formula; The result analysis is clearly presented and supports the conclusion, but it lacks prospects for the future applications.

Author Response

(The authors gave the same response as above.)

Reviewer 4 Report

Thank you for submitting your research in the Eletronics journal. This is an interesting take on the usage of advanced monitoring systems in the field of hydroponics. Here are few of my comments based on the article:

1) The term 'NFT' in the title sounds a bit ambiguous. Kindly change the title without using any acronym.
2) The abstract will make more impact if the authors insert some results in it.
3) Kindly add some more related keywords.
4) The related works are very limited. Kindly put forward more recent studies. Also, a table for the same depicting the advantages and disadvantages which lead to the current study can be highlighted.
4) The 'G-Bell membership function' with 'TDS' and 'pH' membership functions are mentioned. Kindly put some mathematical formulation for the same.
5) As it can be seen from the flowchart of Figure 11, the systems uses a series of 'if-else' conditions; the major question being: Why not use a simple 'decision tree' instead of using 'Neural Networks'? What is the significance of the weights?
6) Figures 7-8 and 9-10 are replicas of each other. Also why is the 'pH' architecture sparse and the 'TDS' dense?
7) Why is the dataset divided into parts? A robust system can be created by merging the datasets with respect to the timestamps of data collection. Please mention the data cleaning process. Some statistical characteristics of the dataset will be more helpful for the readers.
8) The 'Fuzzy Sugeno' takes four fuzzy controllers but only one value is seen in the Table 7. This creates a confusion.
9) Kindly put major results obtained in the 'Conclusion' section. Also the future work need to be focussed.

Hope these comments will help the authors to update the manuscript for its potential publication in the journal.

Author Response

(The authors gave the same response as above.)

Reviewer 5 Report

The topic is interesting and nicely developed. Easy to follow, logical structure.

The 12 references are not enough, I suggest to extend the Introduction section with additional literature on this topic.

In the 13 figure you can see what the Bok Choy plant looks like after 30 days of growing in this system. Is there a control plant that is not grown in this system? What does it look like?

Minor editing mistakes, e.g. figure signatures, sometimes capitalized all the way through, sometimes slightly. Tables are inserted as figure, e.g. 17, 18 figure.

Author Response

(The authors gave the same response as above.)

Round 2

Reviewer 2 Report

The authors reflected the reviewer's comments. The reviewer is still not sure the novelty of this manuscript.

Author Response

Dear Reviewer, 

Thank you for your valuable comment

We have tried our best to revise the paper according to your suggestion.

Please kindly check our reply in attachment file.

Reviewer 3 Report

1) The format of the image naming is not uniform, we should pay attention to whether the image name needs to be bolded and whether there should be a period at the end.

2) In the part of references, there is no indentation in front of the modified references.

Author Response

(The authors gave the same response as above.)

Reviewer 4 Report

The reviewer would like to thank the authors for making the changes in the manuscript. Most of my queries were resolved. I would request the authors to kindly revisit the manuscript again as there are kindly grammatical and sentence formulation mistakes still. In between sentences, there are some capital characters which should not be present.

Overall the work is ready to be excepted once the issues are resolved.

Author Response

(The authors gave the same response as above.)

Round 3

Reviewer 2 Report

The author has revised the introduction to describe the contribution of this manuscript. The reviewer is still not sure how the impact of contribution is, because the methods and the results were not revised.

Author Response

Dear Reviewer, 

Thank you very much for your valuable comment.

We have tried our best to revise our paper to answer your comment. Please kindly check our revision.

Best regards, 

Nico Surantha
